Benchmarking metagenomics classifiers on ancient viral DNA: a simulation study

Arizmendi Cárdenas Yami Ommar 1 2
Neuenschwander Samuel samuel.neuenschwander@unil.ch 1 3
Malaspinas Anna-Sapfo annasapfo.malaspinas@unil.ch 1 2
1 Department of Computational Biology, University of Lausanne , Lausanne , Switzerland
2 Swiss Institute of Bioinformatics , Lausanne , Switzerland
3 Vital-IT, Swiss Institute of Bioinformatics , Lausanne , Switzerland
Gillespie Joseph
Electronic publication date: 2022 Mar 24
Publication date: 2022
Volume: 10
Electronic Location ID: e12784
Received 2021 Apr 12; Accepted 2021 Dec 21
Copyright: ©2022 Arizmendi Cárdenas et al.
Copyright year: 2022
Copyright holder: Arizmendi Cárdenas et al.
License: This is an open access article distributed under the terms of the Creative Commons Attribution License, which permits unrestricted use, distribution, reproduction and adaptation in any medium and for any purpose provided that it is properly attributed. For attribution, the original author(s), title, publication source (PeerJ) and either DOI or URL of the article must be cited.
License URL: https://creativecommons.org/licenses/by/4.0/

Keywords: Ancient DNA, Virome, Classifiers, Simulations, Genomics, Paleomicrobiology, Taxonomic binning, Epidemiology, Sensitivity, Precision

Funding: The Swiss National Science Foundation (SFNS) European Research Council (ERC) The work was supported by the Swiss National Science Foundation (SFNS) and an European Research Council (ERC) grant to Anna-Sapfo Malaspinas. The funders had no role in study design, data collection and analysis, decision to publish, or preparation of the manuscript.

==============================
Owing to technological advances in ancient DNA, it is now possible to sequence viruses from the past to track down their origin and evolution. However, ancient DNA data is considerably more degraded and contaminated than modern data making the identification of ancient viral genomes particularly challenging. Several methods to characterise the modern microbiome (and, within this, the virome) have been developed; in particular, tools that assign sequenced reads to specific taxa in order to characterise the organisms present in a sample of interest. While these existing tools are routinely used in modern data, their performance when applied to ancient microbiome data to screen for ancient viruses remains unknown. In this work, we conducted an extensive simulation study using public viral sequences to establish which tool is the most suitable to screen ancient samples for human DNA viruses. We compared the performance of four widely used classifiers, namely Centrifuge, Kraken2, DIAMOND and MetaPhlAn2, in correctly assigning sequencing reads to the corresponding viruses. To do so, we simulated reads by adding noise typical of ancient DNA to a set of publicly available human DNA viral sequences and to the human genome. We fragmented the DNA into different lengths, added sequencing error and C to T and G to A deamination substitutions at the read termini. Then we measured the resulting sensitivity and precision for all classifiers. Across most simulations, more than 228 out of the 233 simulated viruses were recovered by Centrifuge, Kraken2 and DIAMOND, in contrast to MetaPhlAn2 which recovered only around one third. Overall, Centrifuge and Kraken2 had the best performance with the highest values of sensitivity and precision. We found that deamination damage had little impact on the performance of the classifiers, less than the sequencing error and the length of the reads. Since Centrifuge can handle short reads (in contrast to DIAMOND and Kraken2 with default settings) and since it achieve the highest sensitivity and precision at the species level across all the simulations performed, it is our recommended tool. Regardless of the tool used, our simulations indicate that, for ancient human studies, users should use strict filters to remove all reads of potential human origin. Finally, we recommend that users verify which species are present in the database used, as it might happen that default databases lack sequences for viruses of interest.

Introduction

The human body is home to different species of microorganisms (including bacteria, archaea, viruses and eukaryotes). The composition of these microorganisms is called microbiota, and the union of all their genomes is referred to as the microbiome. The microbiome has been characterised in the last few decades and is known to play an important role in human health (The Integrative HMP (iHMP) Research Network Consortium, 2019). The majority of the microbiome studies have concentrated on the most abundant organisms of the microbiota, namely the bacteria (Stern et al., 2019). However, the rest of the microbiome, especially the virome, the viral fraction of the microbiome, has gained more attention in recent years as it is tightly linked to our welfare (Pérez-Brocal & Moya, 2018; Siqueira et al., 2018; Stern et al., 2019).

Ancient DNA (aDNA) sequencing allows for the reconstruction of ancient microbial genomes and has opened the door to an entire new field sometimes dubbed “paleomicrobiology”. The availability of aDNA has opened a unique window into the past allowing to study the evolution of viruses and other pathogenic microbes (Taubenberger et al., 2005; Duggan et al., 2016; Worobey et al., 2016; Krause-Kyora et al., 2018; Mühlemann et al., 2018a; Mühlemann et al., 2018b; Vågene et al., 2018; Rascovan et al., 2019).

Even though the potential gains are enormous, the ancient microbiome is both challenging to retrieve and to analyse as aDNA is degraded and contaminated. More specifically, aDNA molecules present characteristic patterns caused by post-mortem molecular damage: fragmentation and substitutions (especially occurring at the end of the molecules). Such damage (and the associated patterns) depends on environmental factors such as humidity, temperature, salinity, pH and microbial growth (Briggs et al., 2007; Allentoft et al., 2012; Sawyer et al., 2012; Dabney, Meyer & Pääbo, 2013). Fragmentation is the consequence of depurination with abasic sites leading to single-strand breaks and, subsequently, to double-strand breaks, which cause the DNA molecule to break into small fragments (Dabney, Meyer & Pääbo, 2013). Note that a large fraction of the DNA fragments are considerably shorter than 100 bp resulting in sequenced reads that often include some of the sequencing adapters. Substitutions at the end of the DNA fragments are caused by deamination resulting into an increase of cytosines to thymines (C to T) substitutions at the 5′ end of the fragments and guanines to adenines (G to A) substitutions at the 3′ ends of the fragments for double stranded libraries (Briggs et al., 2007; Carøe et al., 2018).

In most microbiome aDNA studies the first step conducted is to assign a taxon to each of the sequenced reads using a classifier (i.e. classify the reads). The classification of the reads results in a list of microbes that are potentially ancient having infected the host prior to the death of the sampled organism. From this list of potential candidates some microbes (usually pathogens) are selected for follow up analyses to corroborate if they are truly ancient microbes and “endogenous”. The classification of reads and selection of candidates is known as the screening or scanning step (Warinner et al., 2017). The steps following the screening are computationally intensive, for instance, they generally include a mapping step (with BWA (Li & Durbin, 2009) or BowTie2 (Langmead & Salzberg, 2012)) and a blasting step (Altschul et al., 1990) to determine if a particular genome is the most likely source of the sequenced reads (Krause-Kyora et al., 2018; Mühlemann et al., 2018a; Vågene et al., 2018; Mühlemann et al., 2020).

For the screening step there are many different classifiers available. The classifiers are diverse and differ from each other in the underlying database, sequence search algorithm and taxonomic binning (the process of assigning a taxonomic rank to each of the reads sequenced from a sample). In the specific case of ancient microbiome studies, benchmarks have been conducted to choose an appropriate classifier (Velsko et al., 2018) or to determine the most adequate parameters to enhance the performance of specific classifiers (Eisenhofer & Weyrich, 2019). Nevertheless, these benchmark studies only focus on the bacterial fraction of the microbiomes. In this study we have decided to perform a benchmark study to assess the performance of different classifiers in screening ancient microbiome samples for viruses; specifically, our study focuses on DNA viruses with human as their host (human DNA viruses), since most of the ancient viral DNA studies published so far target such viruses. In short, we classified simulated viral data with four classifiers: Centrifuge, Kraken2, DIAMOND and MetaPhlAn2 (for a fifth classifier: MALT, see Supplemental Information 7). The classified samples consisted of simulated ancient-like DNA reads, with different length, deamination damage and sequencing error values. These samples represent an ideal case scenario as they come from a single human DNA viral species and have a high coverage. As we knew the true species in the sample, we scored the classifications per sample; we evaluated the effect of read length, deamination and sequencing error; and we compared the overall performance of all classifiers. In addition, we evaluate the effect of human contamination by classifying simulated human reads with the four classifiers.

The four classifiers assessed in the present study: Centrifuge (Kim et al., 2016), Kraken2 (Wood, Lu & Langmead, 2019), DIAMOND (Buchfink, Xie & Huson, 2015) and MetaPhlAn2 (Truong et al., 2015) were not specifically developed for aDNA. Hence, fragmentation and cytosine deamination may present a challenge for the taxonomic assignment since reads with post-mortem alterations could lead to misclassifications. In the subsequent paragraphs we will describe the characteristics in terms of database, algorithms and taxonomic binning of these four classifiers (see Table 1 for a summary).

Table 1 Characteristics of the five classifiers.

Database, query algorithm and taxonomic binning strategy for each classifier. In the main text the results for MetaPhlAn2 with its custom database and for Centrifuge, Kraken2 and DIAMOND with custom viral databases are shown. The results for Centrifuge and Kraken2’s default databases as well as DIAMOND database with archaeal, bacterial, human and viral protein sequences are shown in Supplemental Information 8. Results for MALT are shown in Supplemental Information 7.

Classifier	Database	Query algorithm	Taxonomic binning strategy	
	               Organisms included	Loci	Molecule	Size			
Centrifuge	Default
(“full db”)	RefSeq: archaea, bacteria,
viruses and human	Whole genomes	DNA	24G	Exact alignment
(no mismatches
or gaps)	Score that favours
longer hits	
Custom
(“viral db”)	RefSeq: viruses	213M	
Kraken2	Default
(“full db”)	Refseq + UniVec_Core:
archaea, bacteria, viruses,
human and (vectors, adapters,
linkers and primers)	Whole genomes	DNA	89G	Exact k-mer matching	Highest number of k-mer
matches considering a
root-to-leaf path	
Custom
(“viral db”)	RefSeq: viruses	34G	
DIAMOND	Custom
(“full db”)	RefSeq: archaea,
bacteria, viruses and human	Protein coding
regions	Proteins	72G	Alignment using
spaced seeds	Lowest common ancestor	
Custom
(“viral db”)	RefSeq: viruses	117M	
MetaPhlAn2	Default	Archaea, bacteria,
viruses and eukaryotic
microbes	Clade specific markers	DNA	1.2G	Alignment with BowTie2: ungapped
seed match followed
by extension	Clade specific
marker	
MALT	Custom
(“viral db”)	RefSeq: viruses	Whole genomes	DNA	7.3G	Banded alignment
with spaced seeds	Lowest common
ancestor	

The four classifiers databases are characterised by the type of molecule and the genomic region(s) that are included. Among the four classifiers, two different types of molecules are used: DNA and protein sequences. In terms of genomic regions, the databases either include multiple loci from each organism (such as a set of proteins or marker genes) or whole genomes. Centrifuge uses whole genome DNA databases. Kraken2 is versatile in database usage as it can use protein or DNA, whole genome or single locus databases. DIAMOND uses a protein database and can use as query both DNA or amino acid sequences. MetaPhlAn2 relies on a multiple loci DNA database consisting of core genes that are shared within a clade but not outside of it (clade specific marker genes). In contrast to its previous version, MetaPhlAn2’s database includes marker genes for viruses and eukaryotic microbes.

For the sequence search algorithm, classifiers either rely on alignments or on exact k-mer matches. Centrifuge, DIAMOND and MetaPhlAn2 depend on alignment algorithms. Centrifuge uses the Burrows-Wheeler Transform (Burrows & Wheeler, 1994) and Ferragina-Manzini index (Ferragina & Manzini, 2000). These algorithms allow to create a data structure which facilitates fast alignments with efficient memory usage. Centrifuge works by searching for exact matches (i.e. no mismatches or gaps are allowed). DIAMOND implements double index alignment (both query and reference are indexed), looking for matches of seeds (short subsequences of fixed length, with default lengths of 15-24 bp) and then extending the alignment. DIAMOND’s seeds are spaced seeds, meaning that some positions in the seeds are treated as wildcards. Finally, MetaPhlAn2 performs alignments using BowTie2 (Langmead & Salzberg, 2012). BowTie2 is an index-assisted aligner that allows gaps in the alignment. Its algorithm has two phases: ungapped seed match and an extension that permits gaps. Kraken2 is in contrast an alignment-free classifier which is based on exact matches of k-mers (sequence substrings of length k) (Wood & Salzberg, 2014). Kraken2 builds an index of k-mers (of length 35 by default for nucleotide sequences) from the database. K-mers are first obtained from the query read and these k-mers are then looked up in the database (an exact k-mer match is performed).

Finally, the classifiers use different strategies for taxonomic binning. DIAMOND uses the lowest common ancestor (LCA) algorithm: that is, if a read matches several species, it will be assigned to the most specific (“lowest”) taxonomic rank shared by all the matching species. If the read only matches one species, the taxonomic assignation will remain at the level of that species. As a consequence, conserved sequences will be assigned to higher taxonomic levels, and specific sequences will be assigned to species level (Huson et al., 2007). Centrifuge and Kraken2 perform taxonomic binning using scoring schemes. Centrifuge ranks the alignments done with a score that favours longer hits. The query read is attributed to the taxon in the database with the highest score (Kim et al., 2016). Similarly, Kraken2 assigns a read to the taxon which has the most k-mer matches in common. In this case, a root-to-leaf path is used to sum k-mers coming from higher taxonomic ranks (Wood & Slazberg, 2014). MetaPhlAn2 relies on a clade specific marker catalogue. If there is an alignment to a marker, the read is assigned to the taxonomic clade associated with that marker. A clade can be as specific as a strain, or as broad as a phylum (Segata et al., 2012).

Materials & Methods

In short, the analyses consisted in simulating reads from reference viral sequences, classifying them with four widely used classification tools and quantifying the performance of the classifiers under different conditions. In particular, we investigated how varying the read lengths, adding DNA damage (features typical of ancient DNA) and sequencing error impact the classifications made by the tools. Finally, we analysed the assignments made by the classifiers on simulated short human reads.

Reference viral sequences

A total of 238 reference viral sequences from 233 different human DNA viruses were used in this study. To get the aforementioned sequences we first queried the Viral-Host DB (Mihara et al., 2016) for viruses that infect Homo sapiens, obtaining a list of 1,315 viruses. We downloaded the reference sequences of such viruses in genbank format from NCBI RefSeq (Brister et al., 2015; O’Leary et al., 2016) and kept the sequences if their “molecule type” record was “DNA” or “ss-DNA”, obtaining a total of 250 DNA viral sequences. Finally, we removed duplicated sequences with the same taxID, representing viruses with different isolates or alternative genomes assemblies; we also removed the Human endogenous retrovirus K113 (HERV-K113) sequence (NC_022518.1), as this virus is a provirus in the human reference genome (Turner et al., 2001). This gave us a total of 233 viruses with 238 different sequences (one of the viruses is stored in RefSeq in six different contigs). A table with information about the viruses used in this study is available on Table S1; it includes the following: virus name, sequence name, sequence length, type of molecule (DNA or ss-DNA), type of genome (linear or circular), sequence accession number, virus taxID and Baltimore group. For each selected viral sequence, we generated a set of simulated reads as described below.

Simulation of viral reads and read length

ART (ART-MountRainier-2016-06-05, art_Illumina Q Version 2.5.8) (Huang et al., 2012) was used to simulate sequencing reads starting from each of the 233 human DNA viruses’ sequences as reference. By default, ART simulates reads with sequencing error typical for the specified sequencer. The sequencing machine was set to Illumina HiSeq 2500 (art_illumina -ss HS25) for all simulations, and the parameter qShift (that controls the amount of sequencing error, see below) was set to 0 for most of the simulations (but also to different values when assessing the effect of sequencing error) resulting in typical Illumina HiSeq 2500 error profiles. In an initial simulation, for each of the 233 viruses, single-end reads of a fixed length were simulated emulating the short read length observed in typical ancient DNA data, see e.g. Green et al. (2008). First, the read length was set at 60 bp (-l 60) and a coverage of 10 × was requested (-f 10) resulting in 280 to 39,270 reads per virus, depending on the viral sequence length. Second, the read length was varied from 30 to 150 bp while keeping the coverage at 10 ×. We set the coverage of 10 × (an unrealistically high number for ancient DNA data) as we wanted to compare the classifiers under ideal conditions. As the reads are classified independently of each other, we do not expect that the relative performance of the classifiers (point estimates) would change with a lower depth of coverage.

Each set of reads from the 233 selected human DNA viruses was classified using four different classifiers: Centrifuge, Kraken2, DIAMOND and MetaPhlAn2 (Table 1). The parameters for each of the classifiers are specified below.

Classification of reads

As much as possible, we used the default parameters suggested by the developers of each tool to build the databases and to perform the classification (as the performance of each tool is likely optimal for those parameters). Otherwise the parameters are stated.

Centrifuge

Centrifuge version 1.0.3-beta was used. To build a custom viral database we downloaded all the viral sequences stored in RefSeq and NCBI’s taxonomy nodes information (files nodes.dmp and names.dmp) (Supplemental File). We created our own “conversion table” file required by Centrifuge which links the accession number to its taxID using the following command of NCBI’s E-utilities (https://dataguide.nlm.nih.gov/eutilities/utilities.html): esearch -db nuccore -query ”${accession}[Accession]” — elink -target taxonomy — efetch -format xml — xtract -pattern Taxon -element TaxId, where “${accession}” represents the accession number. We then created the database with the following command: centrifuge-build –conversion-table my_viral_accession2taxid.map –taxonomy-tree nodes.dmp –name-table names.dmp all_viral.genomic.fna Centrifuge_viral, where “my_viral_accession2taxid.map” is our “conversion-table” file; “nodes.dmp” and “names.dmp” are the taxonomy files downloaded from NCBI Taxonomy; “all_viral.genomic.fna” contains all RefSeq viral sequences in fasta format; and “Centrifuge_viral” is the name of the created database.

We also created Centrifuge’s default database which includes all RefSeq sequences from bacterial, archaeal, viral and human genomes (built on 2 September 2021) following the software instructions). The results obtained with the default database are shown in Supplemental Information 8.

When running Centrifuge to classify the reads a maximum of one taxonomic assignment per read was retrieved (-k 1), to obtain an output comparable to the other classifiers. All other parameters were left with default values.

Kraken2

Kraken2 version 2.0.7-beta was run with default parameters. To build our viral custom database first we downloaded the taxonomy using the following command kraken2-build –download-taxonomy –db Kraken2_viral_db, where Kraken2_viral_db is the name of the database. Then we replace all the internal files (except “Kraken2_viral_db/taxonomy”) to have the following directory structure:

Kraken2_viral_db/

—

—– library

—\– viral

—\– library.fna

—

—– seqid2taxid.map

—

\– taxonomy where “library.fna” is the same as “all_viral.genomic.fna” and “seqid2taxid.map” is the same as “my_viral_accession2taxid.map”; both files used to build Centrifuge’s custom viral database. To build the database the following command was used: kraken2-build –build –db Kraken2_viral_db.

We also downloaded Kraken2 prebuilt “standard” database from https://benlangmead.github.io/aws-indexes/k2 dated to 17 May 2021. This database includes the RefSeq of bacterial, archaeal, viral and human genomes plus UniVec_Core. The results obtained with this default database are shown in Supplemental Information 8, we also performed analyses building a database with a shorter k-mer length in order to classify short (30 bp) reads, for these results see Supplemental Information 9.

DIAMOND

DIAMOND version 0.9.22 was used with the RefSeq viral protein database. The protein sequences, the taxonomy nodes, and the file which links accession number to taxID were downloaded from NCBI (Supplemental File). All files were downloaded on 1 September 2021. The database was built following the manual’s instructions.

In addition, we built a “full” database with archaeal, bacterial, human and viral protein sequences. The viral sequences and taxonomy files are identical as the previous database. We downloaded the archaeal sequences, the bacterial, and human from NCBI (Supplemental File). Archaeal and bacterial protein sequences were downloaded on 1 September 2021; human protein sequences were downloaded on 6 September 2021. The results obtained with this “full” database are shown in Supplemental Information 8.

DIAMOND was run in the taxonomic classification mode (-f 102 option) in order to perform the taxonomic assignments; all other parameters were set as default. As the input data is DNA, the query is translated into protein for each of its six reading frames and aligned against the protein database.

We also performed analyses changing the default seed length used by DIAMOND in order to classifiy short (30 bp) reads, for these results see Supplemental Information 9.

MetaPhlAn2

MetaPhlAn2 version 2.6.0 was run with default parameters. The default microbe clade-specific marker genes database was used (mpa_v20_m200 version), it includes marker genes from bacteria, archaea, viruses and eukaryotic microbes.

Summary statistics

To evaluate the performance of the classifiers, each read was assigned to one of the following categories: i) correctly classified read at the species level (“correct species”), ii) correctly classified read at a higher taxa level (“correct higher”), iii) misclassified read at any taxa level (“incorrect”) and iv) unclassified read (“unclassified”). Figure 1 shows a schematic example of each of the four categories for Variola virus. The correctly classified reads are divided into two sets hereafter: a set including only the reads classified correctly at the species level (“correct species”, abbreviated “s” below) and a set including all the reads classified correctly at the species and at higher taxa (“correct species” and “correct higher”, abbreviated “s&h” below). Further, we computed the widely used statistics sensitivity and precision to compare the classifiers (Wood & Salzberg, 2014). Sensitivity is the proportion of correctly classified reads over all simulated reads; precision is the proportion of correctly classified reads over all classified reads. We computed two sets of sensitivity and precision measures depending on whether “correct species” is assumed to be correct or “correct species” and “correct higher” are assumed to be correct. More specifically, sensitivity and precision are defined as follows for each viral sequence vi:

Figure 1 Visual representation of the different classification categories defined in this study (“correct species”, “correct higher”, “incorrect” and “unclassified”).

The Variola virus is used here as an example to illustrate the simulation steps and the four classification categories used to summarize the simulation results. Three schematic steps are shown to summarize the simulations: (1) The selection of a virus (here the Variola virus). (2) The simulation of reads from the viral sequence: the selected virus is used as reference genome to simulate sequencing reads with ART (Huang et al., 2012). (3a) The classification of the reads: all reads are then classified with Centrifuge, Kraken2, DIAMOND and MetaPhlan2 (Buchfink, Xie & Huson, 2015; Truong et al., 2015; Kim et al., 2016; Wood, Lu & Langmead, 2019). And then assigned to one of the four categories (3b): i) “correct species”: reads correctly classified as the virus of interest (dark green), ii) “correct higher”: reads classified as a higher taxonomic rank included in the lineage of the virus of interest (light green), iii) “incorrect”: reads classified as a taxon not included in the lineage of the virus of interest, iv) “unclassified”: reads not classified.

Sensitivitysvi=rcsvircsvi+rchvi+rivi+ruvi

Sensitivitys&hvi=rcsvi+rchvircsvi+rchvi+rivi+ruvi

Precisionsvi=rcsvircsvi+rchvi+rivi

Precisions&hvi=rcsvi+rchvircsvi+rchvi+rivi

where rcsvi are the number of “correct species” reads for the viral sequence vi, rchvi are the number of “correct higher” reads for the viral sequence vi, rivi are the number of “incorrect” reads for the viral sequence vi, ruvi are the number of “unclassified” reads for the viral sequence vi. For some of the viral sequences vj, all reads are “unclassified” and rcsvj+rchvj+rivj=0. In this case, the Precisionsvj and the Precisions&hvj are undefined.

We summarize the results by computing the mean sensitivities and precisions across the simulated viral sequences. For the sensitivities, we have:

Sensitivitys=1233∑i=1233rcsvircsvi+rchvi+rivi+ruvi

Sensitivitys&h=1233∑i=1233rcsvi+rchvircsvi+rchvi+rivi+ruvi

For the precisions, we have:

Precisions=1nc∑i=1ncrcsvircsvi+rchvi+rivi

Precisions&h=1nc∑i=1ncrcsvi+rchvircsvi+rchvi+rivi

where nc is the number of viral sequences for which there is at least one read classified (correctly or not), i.e. for which rcsvi+rchvi+rivi>0.

In addition, we counted and characterised the number of spurious extra taxa identified by the classifiers. These taxa are associated with one or more “incorrect” reads. They include any taxa that were not used to simulate the sequencing reads or taxa not included in the lineages of the simulated viruses.

Effect of adding deamination damage

Besides DNA fragmentation, another molecular characteristic of ancient DNA is the deamination process which, for double stranded libraries, results in an increase of C to T substitutions at the 5′ ends and G to A substitutions at the 3′ ends (Briggs et al., 2007; Carøe et al., 2018). We simulated reads of 60 bp length and added different levels of deamination using deamSim (June 06 2016 version) gargammel sub-program (Renaud et al., 2017). The following parameters were applied: a nick frequency of 0.03, a geometric parameter of 0.25 for the average length of overhanging ends, a probability of deamination in the double-stranded portions of DNA of 0.01, and 11 different values for the probability of deamination in the single stranded portions of the DNA (0, 0.05, 0.1, 0.15, 0.2, 0.25, 0.3, 0.35, 0.4, 0.45 and 0.5). Note that the resulting damage pattern is similar to what has been observed for real data (Fig. 2) with higher deamination fractions at read termini. The resulting observed average number of substitutions is shown in Fig. 3 for NC_006273.2 (Human betaherpes virus 5) for all tested single-stranded probabilities of deamination.

Figure 2 Observed deamination damage across the reads.

Observed frequency of substitutions across the reads simulated from the viral sequence NC_006273.2 (Human betaherpes virus 5) for different probabilities of single stranded deamination ranging from 0 to 0.5 (the range used in the simulations). The nick frequency was set at 0.03, the geometric parameter for the average length of overhanging ends at 0.25 and the probability of deamination in the double-stranded portions of DNA at 0.01. The frequency of substitutions is shown on the y-axis; the distance from the read termini is shown on the x-axis. C to T substitutions are depicted in orange; G to A substitutions are depicted in blue; in grey all other substitutions. Plots on the left represent the 5′ end of the reads, plots on the right represent the 3′ end. Note that similar patterns have been observed in real data (Briggs et al., 2007; Carøe et al., 2018).

Figure 3 Average number of substitutions for the deamination and sequencing error simulations.

Observed average number of substitutions for different simulation parameter values for the deamination and sequencing error simulations. Shown here are the observed values for the case of NC_006273.2 (Human betaherpes virus 5) with 39,270 simulated reads of length 60 bp. For the deamination simulations, the single-stranded probability (see Fig. 2) of the deamSim gargamel subprogram (Renaud et al., 2017) is increased from 0 to 0.5. Note that deamination takes place with a probability of 0.01 across the read for all the deamination simulations so that there are additional errors even with a single stranded probability set at 0. For the sequencing error simulations, the parameter qShift of ART was decreased from 0 to −9 which corresponds to a 1 fold (qShift of 0) to 7.9 fold (qShift of −9) increase compared to standard Illumina HiSeq2500 sequencing (Huang et al., 2012).

Effect of increasing substitution sequencing error

To better understand the effect of the distribution of errors across the reads, substitution errors were also simulated following an Illumina-like error profile. To do so, sequencing errors were added to the simulated reads using ART (Huang et al., 2012) indirectly varying the overall error rate by changing the parameter qShift of ART. Specifically, seven different values for the parameter qShift were tested (0, −1, −3, −5, −7 and −9). This parameter changes the quality score of the simulated reads, leading to more substitutions in the simulated reads with lower values. Each qShift value corresponds to an overall fold increase of substitution error (Huang et al., 2012):

fold_increase= 110qShift10

For instance, a qShift of 0 corresponds to a one-fold increase (i.e. no increase or in other words typical sequencing error), −3 to a ∼two-fold increase and, finally, −9 to a 7.9-fold increase. The effect of decreasing qShift on the observed number of substitutions in simulated reads is shown in Fig. 3 for the case of NC_006273.2 (Human betaherpes virus 5) values. For this analysis also the simulated read length was set at 60 bp.

Human simulations

When human remains are sequenced, the resulting data contains microbial and human DNA. To study the microbiome, ideally, reads mapping to the human genomes should be first removed. However, such cleaning steps are imperfect and can lead to false classifications. To evaluate how the tools classify potential human sequences that were not properly filtered out, human reads were simulated using ART (ART-MountRainier-2016-06-05, art_Illumina Q Version 2.5.8) with the human genome (GRCh37) as reference. As above, the read length used was 60 bp, the coverage 10 ×, and the sequencing technology was set to Illumina HiSeq 2500. Centrifuge, Kraken2, DIAMOND and MetaPhlAn2 were then used to classify the reads as above. The proportions and the number of correctly (i.e. classified as human) and incorrectly (i.e. not classified as human or as a taxon within the human lineage) classified reads were calculated; the number of extra taxa were counted and characterised by determining the assigned kingdom and the number of assigned reads per taxon.

Results

Overall, the performance of the four classifiers is surprisingly distinct across simulations (Figs. 4-9). The results for Centrifuge and Kraken2 are the most similar to each other, MetaPhlAn2 is the most distinct compared to Centrifuge and Kraken2, while DIAMOND occupies an intermediate position.

Figure 4 Classification results for the 60 bp read set.

(A) Mean sensitivity versus mean precision. The mean sensitivities (Sensitivity_s & Sensitivity_s&h) are the means of the proportions of reads correctly classified over the total number of simulated reads across viruses. The mean precisions (Precision_s & Precision_s&h) are the means of the proportions of reads correctly classified over the number of classified reads across viruses. Circles denote the values if only “correct species” reads are considered as correctly classified reads; triangles denote the values if “correct species” and “correct higher” reads are considered as correctly classified reads (see Materials and Methods). The perfect classifier would have 100% sensitivity and 100% precision. (B) Total number of viruses recovered for each classifier when correctly identifying at least 1 read per virus. The dashed line indicates the total number of tested viruses (233). (C) Mean number of spurious extra taxa per classifier. In this plot, a taxon is assumed as identified by a classifier if at least 1 read is assigned to it.

Figure 5 Effect of the read length on the classification performance.

For these simulations, read length was varied from 30 to 150 bp. (A) Average Sensitivity_s (continuous lines) and Sensitivity_s&h (dashed lines) for each classifier. (B) Average Precision_s (continuous lines) and Precision_s&h (dashed lines) for each classifier. (C) Total number of viruses detected out of the 233 tested. The dashed line shows the maximum number of detectable viruses. (D) Average number of spurious extra taxa across simulated viral sequences. The vertical dashed line indicates the initial 60 bp read set.

Classification results for the 60 bp read set

Figure 4 and Fig. S1 summarise the results for the simulated 60 bp read set. As discussed above, the simulated reads could be either correctly classified at the species level (“correct species”), correctly classified at a higher taxa level (“correct higher”), misclassified (“incorrect”) or “unclassified” (Fig. 1).

Figure 6 Effect of the deamination damage on the classification performance.

For these simulations, errors were added using deamSim gargamel subprogram which assumes an ancient DNA deamination-like distribution and were added in addition to the ART Illumina like sequencing errors. The results shown here correspond to a single-stranded probability of deamination varying from 0 to 0.5. For all the results the nick frequency is set at 0.03, the average length of overhanging ends is set at 0.25, and the probability of deamination in the double-stranded portions of DNA is set at 0.01. A) Average Sensitivity_s (continuous lines) and Sensitivity_s&h (dashed lines) for each classifier. (B) Average Precision_s (continuous lines) and Precision_s&h (dashed lines) for each classifier. (C) Total number of viruses detected out of the 233 tested. The dashed line shows the maximum number of detectable viruses. (D) Average number of spurious extra taxa across simulated viruses.

When considering each virus separately, most reads are classified correctly at the species level (“correct species”) for most of the simulated viral sequences for Centrifuge and Kraken2 (Fig. S1). In contrast, for DIAMOND, the proportion of “correct species” varies substantially with many viruses with low, intermediate and high “correct species” proportions while the remaining reads are mostly “unclassified” or “correct higher”. MetaPhlAn2 stands out by having roughly two thirds of the viruses with no correctly classified reads (neither as “correct species” nor as “correct higher”). As a result, the means across viruses of the proportion of “correct species” are 24.41% (MetaPhlAn2), 53.61% (DIAMOND), 93.39% (Kraken2) and 94.48% (Centrifuge). Unlike the other classifiers, DIAMOND classifies a large proportion of reads correctly but at a higher taxonomic rank (“correct higher”) for many of the viruses. The other three classifiers have a handful of viruses with a large proportion of “correct higher”. Overall, most of the simulated viral sequences have zero or a small proportion of “incorrect” reads, especially in the case of Centrifuge and Kraken2 (Fig. S1, for detailed list of the viruses with the highest incorrect proportions per classifiers see Supplemental Information 10; Table S2 contains the raw numbers of reads assigned to each taxa per classifier). The resulting averages of the proportions of “incorrect” reads are 0.01% (Centrifuge), 0.26% (Kraken2), 0.75% (DIAMOND) and 2.79% (MetaPhlAn2). Finally, Centrifuge and Kraken2 classify most of the simulated reads for all viruses (with a mean “unclassified” of 0.01% and 0.44%), while both DIAMOND (mean of 20.4%) and MetaPhlAn2 (mean of 59.8%) do not classify a large proportion of reads for most viruses (Fig. S2).

Figure 7 Effect of the substitution sequencing error on the classification performance.

For these simulations, errors were added using ART which assumes a profile similar to the ones observed for Illumina Sequencing machines (HiSeq 2500). The results shown here correspond to increasing the overall sequencing error rate ranging from 1 to 7.9-fold (qShift values from 0 to −9). On the x-axis, the first number correspond to the expected fold increase in error rate while the parameter that was varied, qShift, is shown in parenthesis. (A) Average Sensitivity_s (continuous lines) and Sensitivity_s&h (dashed lines) for each classifier. (B) Average Precision_s (continuous lines) and Precision_s&h (dashed lines) for each classifier. (C) Total number of viruses detected out of the 233 tested. The dashed line shows the maximum number of detectable viruses. (D) Average number of spurious extra taxa across simulated viral sequences. The vertical dashed line indicates the initial 60 bp read set.

The sensitivity and precision reflect the previous “correct species”, “correct higher”, “incorrect” and “unclassified” overall values (Fig. 4A circles and triangles). At the species level, both means are considerably higher for Centrifuge and Kraken2 with mean values above 93% for Sensitivity_s and Precision_s. In contrast, MetaPhlAn2’s sensitivity and precision are the lowest with values of 24.4% (Sensitivity_s) and 46.3% (Precision_s). MetaPhlAn2’s low Sensitivity_s value is a consequence of the high proportions of “unclassified” reads for this classifier. Note that for this classifier, the precision is computed by excluding the viruses without any classified reads, i.e. by excluding most of the data (see Material and Methods). At the species level, DIAMOND has intermediate values, with a Sensitivity_s of 53.6% and a Precision_s of 67.4%; both lower than Centrifuge and Kraken2’s, but higher than MetaPhlAn2’s Fig. 4A circles).

When considering the “correct species” and “correct higher” reads to compute the precision and the sensitivity (Fig. 4A triangles), the order of the classifiers stays the same, however the performance of DIAMOND and MetaPhlAn2 are greatly improved. In particular, DIAMOND has many viruses with a large fraction of “correct higher” reads. As a result, DIAMOND’s Sensitivity_s&h is considerably higher than its Sensitivity_s with values of 78.9% and 53.6%, respectively. Similarly, its Precision_s&h (98.6%) is higher than its Precision_s (67.4%). DIAMOND’s Sensitivity_s&h is lower than Centrifuge’s and Kraken2’s, and higher than MetaPhlAn2’s. Centrifuge and Kraken2 achieve the highest Precision_s&h values with 99.9% and 99.7%, respectively; followed closely by DIAMOND (98.6%), and lastly by MetaPhlAn2 (74.4%) (Fig. 4A triangles).

Beyond the proportion of correctly classified reads per virus, we also considered which viruses used in the simulations were recovered with at least one correctly classified read (Fig. 4B). Encouragingly, despite large differences observed at the read level, three classifiers detected almost all the 233 viruses tested: Centrifuge and Kraken2 identified 232 viruses, and DIAMOND detected 228. In contrast MetaPhlAn2 reported only 87 viruses.

In addition, we computed how many spurious extra taxa were reported, these are taxa that were neither the virus of interest nor in the lineage of that virus (Fig. 4C). No classifier reports more than two extra taxa per virus on average. We found that MetaPhlAn2 is the classifier with the highest number of spurious extra taxa (on average 1.45 across tested viruses), followed by DIAMOND (1.06), Kraken2 (0.58) and, finally, Centrifuge (0.41).

Effect of increasing read length on the classification performance

To better understand the effect of the length of the sequenced reads on the classification results, we then investigated the performance of the classifiers for read lengths ranging from 30 to 150 bp (Fig. 5, Figs. S2 & S3).

In general, by increasing the read length, the classifications improve. We observe higher proportions of “correct species” reads (Figs. S2 & S3), fewer “incorrect” reads, higher sensitivity and precision, and a decrease in the number of spurious extra taxa (Figs. S2 & S3).

The most drastic result across classifiers for the read length simulations is that DIAMOND and Kraken2 do not classify very short reads. Specifically, no 30 bp reads are classified for DIAMOND and for Kraken2 given the default parameters used. Given these results, we have changed their default settings to classify these short 30 bp reads, however we found that the number of classified reads was minimal for DIAMOND. In the case of Kraken2, less than half of the reads are classified and an increase of spurious extra taxa are reported, the results are shown in Supplemental Information 9. Similarly, DIAMOND classifies only very few reads for 40 bp and 50 bp compared to higher read lengths. This contrasts with the results for Centrifuge and MetaPhlAn2 whose performance do not significantly deteriorate with shorter reads in the case of Centrifuge, or remains stable across all lengths in the case of MetaPhlAn2 (Fig. 5).

From 60 bp onwards, most of the observations discussed above hold; Centrifuge and Kraken2 have the highest “correct species” mean proportions, followed by DIAMOND and MetaPhlAn2 (Fig. S3). DIAMOND has the highest “correct higher” proportions; and Centrifuge has the lowest “incorrect” mean proportions. Consequently, from 60 bp onwards, the highest sensitivities (Sensitivity_s) and precisions (Precision_s) are observed for Centrifuge and Kraken2 followed by DIAMOND and MetaPhlAn2. Regarding the “correct higher” classifications, Centrifuge, Kraken2 and DIAMOND have the highest Precision_s&h; as DIAMOND classifies correctly many reads at higher taxonomic level its precision is greatly benefited. DIAMOND’s has a Sensitivity_s&h that is lower than Centrifuge’s and Kraken2’s but higher than MetaPhlAn2. Moreover, as above, regardless of the simulated read length, the number of detected viruses remains the same. Most viruses are detected by Centrifuge, Kraken2 and DIAMOND while an important fraction of the tested viruses are missed by MetaPhlAn2 (Fig. 5C).

A sharp decrease in the number of extra taxa for increased read lengths is observed for MetaPhlAn2 (Fig. 5D), the most constant classifier across read lengths for sensitivity. A slighter decrease of spurious extra taxa is observed for Centrifuge and Kraken2. In the case of DIAMOND, we see an increase up to 90 bp, followed by a decrease. The highest number of spurious extra taxa is found in MetaPhlAn2 up to 60 bp; for read lengths above 60 bp DIAMOND present the highest number (Fig. 5D).

When looking more in detail per classifier, for read lengths of 40 bp and longer, the proportion of “correct species” reads increases for Centrifuge and Kraken2, while the proportions of “correct higher” and “incorrect” reads decreases (Figs. S2 & S3). This results in a slight increase in the sensitivity and precision with longer reads (Figs. 5A & 5B). For DIAMOND, the fraction of “unclassified” reads decreases dramatically from 100% (30 bp read lengths) to less than 25% (150 bp read lengths). Among the reads that are classified, the “correct species” and “correct higher” ones increase substantially up to 90 bp while the proportion of “correct higher” slightly decreases above 90 bp (Figs. S2 & S3). Unlike other classifiers, substantially more reads are classified with longer read lengths, and a slight increase in the proportion of “incorrect” reads is observed as well (Figs. S2 & S3). As a result, the sensitivity improves substantially for longer reads for DIAMOND, especially for the Sensitivity_s&h while the precision’s increase is slighter (Fig. 5).

MetaPhlAn2 is the most stable classifier when varying read length with the classification categories “correct species” and “correct higher” remaining almost constant (Figs. S2 & S3). The only summary statistics that changes is the proportion of “incorrect” reads which slightly decrease when increasing read length, resulting also in a decrease in the number of spurious extra taxa. Consequently, the sensitivity remains constant and the precision increases with longer reads (Figs. 5A & 5B).

Effect of the deamination damage on the classification performance

To further assess the performance of the classifiers for ancient DNA like data, we added deamination damage characteristic of ancient double-stranded DNA libraries (increase of C to T substitutions at the 5′ read termini and G to A substitutions at the 3′ read termini, see Fig. 2) to the simulated 60 bp reads and evaluated the impact of deamination on the taxonomic assignments (Fig. 6). The probability of deamination in the double-stranded portions of DNA was set at 0.01 for all simulations, while we varied the different probabilities of single stranded deamination from 0 to 0.5 (see Material and Methods). Those parameters result in realistic damage profiles, with 3′ values of up to 0.45 being on the higher end of what can be observed (Fig. 3), see e.g. Malaspinas et al. (2014) and Allentoft et al. (2015).

Overall, reassuringly, deamination has little to no effect on the classifiers’ performance. The slight reduction of “correct species” reads leads to a hardly noticeable decrease of the sensitivities (Fig. 6A). Similarly, the precision of all classifiers remains constant for all values of the single-stranded deamination probabilities (Fig. 6B). The deamination damage has no effect, as well, in the number of correctly detected viruses (among the 233 viruses tested) (Fig. 6C).

Deamination only impacts the average number of spurious extra taxa for Centrifuge, Kraken2 and DIAMOND, which increases (Fig. 6D). In contrast, for MetaPhlAn2 the average number of spurious extra taxa presents a small reduction. For single-stranded probability of deamination values ranging from 0 to 0.15 Centrifuge reports the lowest number of spurious extra taxa; from 0.2 and above DIAMOND has the lowest numbers. MetaPhlAn2 reports the highest number of spurious extra taxa up to 0.35; above that value Centrifuge presents higher numbers.

Figure 8 Classification of human reads.

(A) Proportion of reads for each classification category (see Fig. 1). Note that none of the databases used include human (see Table 1). (B) Total number of incorrectly classified reads. (C) Number of spurious extra taxa reported by each classifier.

Effect of the substitution sequencing error on the classification performance

As the effects of ancient DNA deamination were minor, we indirectly investigated whether this was the result of the unusual ancient DNA-like distribution and type (C to T in 5′ ends and G to A in 3′ ends) of the substitutions compared to standard sequencing error. For these simulations, errors were added using ART which assumes a profile similar to the ones observed for Illumina Sequencing machines (HiSeq 2500) by decreasing the qShift parameter resulting in an increase of 1 to 7.9-fold of the overall error rate. As with damage, a higher number of errors leads to divergent reads (reads with different sequences compared to the reference from which they were generated, Fig. 3).

As for damage, DIAMOND and MetaPhlAn2 are essentially not affected by the sequencing error across simulations with results for a 7.9-fold increase in error rate essentially identical to the ones at 1-fold across all statistics (Fig. 7, Figs. S2 & S3).

Figure 9 Taxonomic ranks represented in the classification of human reads.

Number of reads and the proportion of reads per classifier (bar plots). Proportion of taxa classified as a particular pair of superkingdom-kingdom (pie charts).

The trends observed for Centrifuge and Kraken2 (increase of “unclassified” reads and small decrease of the rest of the categories) are exacerbated with Illumina-like sequencing error compared to damage (Figs. S2 & S3). This result is an almost linear decrease in sensitivity when compared to the fold increase in error for Centrifuge and Kraken2 (Fig. 7). However, Centrifuge and Kraken2 precision values remain constant, the reason for this is that the absolute number of correctly classified (“correct species” and “correct higher”) is high relative to “incorrect” along the increased substitutions (Fig. 7B).

When considering not only the proportion of reads but also the viruses identified, we observe that the number of recovered true viruses is constant regardless of the error rate substitutions. Moreover, as for deamination, the number of spurious extra taxa increases substantially for Centrifuge and Kraken2 with increased number of substitutions, while DIAMOND’s increase is less sharp and MetaPhlAn2 presents an overall small reduction (Fig. 7D).

Classification of human reads

In recent years, many ancient microbes have been detected in ancient human remains. In those studies, one of the challenges is to disentangle human from microbial DNA. To assess the effect of not filtering out effectively human DNA, simulated reads were generated from the reference human genome and used as input to the classifiers.

Almost the totality of reads (more than 98%) remains unclassified for all four classifiers (Fig. 8A), which is expected as none of them have human sequences in the databases used. MetaPhlAn2 and DIAMOND have the highest proportion of unclassified reads (99.98%), followed by Kraken2 (99.5%) and Centrifuge (98.4%). Among all classifiers, Centrifuge presents the highest number of incorrectly assigned reads (Fig. 8B). Above 78% of the classifications were made at the species level for all classifiers; and about a quarter of the assignments were within the Heunggongvirae kingdom (Fig. 9).

Interestingly, the number of spurious extra taxa varied greatly across classifiers (Fig. 8C), being Centrifuge the tool with the largest number of spurious extra taxa with 7,118 identified by at least 1 read, followed by Kraken2 (1,390), DIAMOND (528) and MetaPhlan2 (91). Table S3 contains a list of each taxon identified by at least one read for each classifier, we observed that several taxa have a large number of assigned reads (several thousands) for all classifiers.

Discussion

By providing a direct window into the past, ancient virome studies have the potential to shed light into the pathogens responsible for historical epidemics, to uncover prehistorical epidemics, but also to provide clues about the molecular biology and the evolutionary history of ancient viruses. Yet, finding ancient viruses is akin “finding broken needles in noisy haystacks”. Ancient genomes are fragmented, affected by post-mortem damage, incomplete and contaminated by ancient and present organisms. Furthermore, the DNA of viruses represent only a tiny fraction of the DNA extracted from the host. Thus, the recovery of ancient viruses’ genomes is an experimental and computational challenge and, given how rare ancient samples are, it is crucial to recover as much ancient DNA as possible.

Our approach and its limitations

To get a sense of the best suited classifier to screen for viruses in ancient microbiome samples, we compared state-of-the art classifiers under controlled conditions, i.e. in silico simulations. To do so, we selected 238 viral sequences from 233 human DNA viruses, fragmented them and added ancient DNA-like noise, and classified the resulting short and damaged reads with Centrifuge, Kraken2, DIAMOND and MetaPhlAn2. These classifiers (that were not specifically developed for ancient DNA) differ in essentially all possible ways including the underlying databases, the sequence search algorithms and the taxonomic binning strategies (Table 1). Hereafter we provide some clues to explain the main results that were obtained for each classifier and conclude with some recommendations. However, it is important to note that our results and recommendations are limited to the four tested classifiers and to the databases available at the time of the analyses. Moreover, the simulations are a proxy of a real-life situations (i.e. a screening step in a microbiome study). In other words, our recommendations are a good starting point for anyone wishing to study ancient viromes. Yet future work should allow to update and to further refine them.

Unidentified and incorrectly classified viruses

In most simulations with reads over 50 bp long, three of the classifiers, namely Centrifuge, Kraken2 and DIAMOND, successfully detect at the species level more than 228 out of the 233 tested viruses. These encouraging results suggests that if viral genomes were present in a sample with sufficient coverage, they would be detected by Centrifuge, Kraken2 and DIAMOND even if the reads are fragmented and damaged. In contrast, MetaPhlAn2 missed 146 viruses. Regarding the incorrect assignments, encouragingly, for almost all viruses across all classifiers (except for MetaPhlAn2), the incorrect proportion does not exceed 43%. For a list as well as a discussion of the unidentified viruses and viruses with the highest incorrect proportions see Supplemental Information 10.

Classification of very short (30 bp) reads

One of the main characteristics of ancient DNA is the highly fragmented nature of the molecules, generally shorter than the sequencing read length (resulting in the sequencing of the adapters). The classifiers we tested were not developed to tackle such short reads and as expected, longer reads, which contain more information, positively impact the overall classification results. Our simulations suggest that DIAMOND and Kraken2’s performance is substantially affected by a reduction in read length. This contrasts with the results for Centrifuge and especially MetaPhlAn2, showing these two classifiers can still handle shorter reads. In particular, we observed that 30 bp long reads are not classified at all by DIAMOND and Kraken2 using their default parameters. DIAMOND’s performance is still substantially reduced at 40 bp and 50 bp while Kraken2’s sensitivity is close to its highest value at 40 bp. In other words, with default parameters, a significant number of reads shorter than 40 bp would be lost when using Kraken2; and in the case of DIAMOND, shorter than 60 bp. These results can be explained by the underlying algorithms and databases. Kraken2’s default k-mer length (35 bp) was used to build its database, making an exact match impossible for 30 bp, as the k-mers are longer than the reads. Similarly, DIAMOND uses as seeds 4 shapes of length 15–24 and weight 12 by default which translates into 45–72 bp and weight 36. This explains why it failed to classify any read of 30 bp long, and why it has issues classifying 40 bp reads (the reads being only 4 bp longer than the default seed weight). Relatedly, DIAMOND is the only classifier depending on protein alignments and longer DNA reads are required to align the same number of amino acids as nucleotides. Even when changing the default settings of DIAMOND and Kraken2 the number of short reads classified is minimal (DIAMOND) or it comes with a trade-off: higher number of spurious taxa reported (Kraken2), see Supplemental Information 9.

A high sensitivity and precision for Centrifuge and Kraken2

As aDNA studies are usually limited in the amount of starting biological material, it is crucial to assign correctly as many reads as possible and to minimise the number of errors, i.e. to use a classifier with a high sensitivity and precision. With sensitivity and precision values at the species level around 90% and above, Centrifuge and Kraken2 outperform DIAMOND and MetaPhlAn2. In other words, Centrifuge and Kraken2 classify correctly more reads among the simulated reads but also classify correctly more reads among the classified ones. In comparison, DIAMOND and MetaPhlAn2 detect correctly less than 65% of the reads (sensitivity) on average across viruses (Fig. 4A). For the sensitivity, this can likely be explained in part by the differences in database; Centrifuge and Kraken2 include whole genomes, while DIAMOND’s database contains proteins and MetaPhlAn2’s contains custom clade specific markers, i.e. only a fraction of the genomes in both cases. The latter two classifiers have therefore a large proportion of unclassified reads. DIAMOND’s precision is not higher at the species level as it classifies correctly a large fraction of reads at higher taxonomic ranks (“correct higher”, see below). MetaPhlAn2’s mean precision is considerably lower than the other classifiers because it has high proportions of “incorrect” reads for some viruses, and because the number of viruses used to compute is smaller (only the viruses with classifications are considered, and many viruses had 0 classified reads). As mentioned in Supplemental Information 10, three and 145 out of the 233 tested viruses do not have sequences stored in DIAMOND’s and MetaPhlAn2’s databases, respectively. As a result, both sensitivity and precision are reduced for both tools. When computing the statistics considering only the viruses that are present in their databases their performance improves: with 60 bp long reads DIAMOND reaches a Sensitivity_s of 54.31%, Sensitivity_s&h of 79.58%, Precision_s of 68.24% and Precision_s&h of 99.24%; while MetaPhlAn2 reaches a Sensitivity_s of 65.37%, a Sensitivity_s&h of 69.67%, a Precision_s of 89.9% and a Precision_s&h of 96.92%. Despite this improvement both of them are still outperformed by Centrifuge and Kraken2.

High proportions of “correct higher” reads for DIAMOND

DIAMOND exhibits a markedly higher number of reads correctly classified at a higher taxonomic rank compared to the other classifiers. Protein changes have the potential to directly impact the phenotype of an organism and are therefore more conserved (Li, 1997) and DIAMOND is the only classifier based on a protein database. Thus, the protein database (which benefits from the redundancy of the genetic code and the high degree of conservation of proteins) but also the LCA binning algorithm implemented in DIAMOND may explain why this classifier has a much larger fraction of reads classified as “correct higher”, and as a result a great improvement of its precision when considering the higher taxa. As an example, the virus with the highest proportion of reads assigned to higher levels in DIAMOND, human erythrovirus V9 (NC_004295.1) has 84.8% of its reads correctly assigned to the Erythroparvovirus genus. All these reads aligned to proteins of the correct species human erythrovirus V9, but also to proteins from species within the genus. Given the use of the LCA algorithm, DIAMOND reported the reads at genus level. Consequently, when using DIAMOND, one should consider jointly the classifications at the species level and above.

Sensitivity and precision are generally robust to deamination damage

Besides fragmentation, deamination damage is another key feature of ancient DNA that could negatively impact classifications. Encouragingly, our simulations suggest that ancient DNA-like damage has little to no effect on the classifiers’ performance aside for an increased number of spurious extra taxa for Centrifuge, MetaPhlAn2 and Kraken2 (see below). To avoid a high number of spurious extra taxa, the use of DIAMOND would be convenient, as long as the reads are long enough. Interestingly, when comparing deamination and sequencing error simulations resulting in similar number of mismatches (i.e. when the single stranded probability of deamination is set at 0.5 and the qShift value at −9), deamination has little effect, while sequencing error reduces the sensitivities of Centrifuge and Kraken2. This suggests having errors mostly concentrated at the end of the reads (ancient DNA-like errors) do not hinder their classification as much as Illumina-like errors, which are any kind of substitutions more evenly distributed across the read (Pfeiffer et al., 2018).

Extra spurious taxa

Across all simulations with reads above 30 bp, Centrifuge and Kraken2 achieved the lowest fraction of misclassified reads across viruses and MetaPhlAn2 the highest. Besides the proportion of “incorrect” reads, we also investigated the number of spurious extra taxa reported since they provide a clue of the amount of follow-up work to confirm candidate microbes. In length simulations, MetaPhlAn2 and DIAMOND have the highest number of spurious extra taxa. In contrast, Centrifuge is the classifier with the lowest number of spurious taxa for the same simulations; as the length increases the number of spurious extra taxa decreases. This could be explained by Centrifuge’s taxonomic binning strategy with scores favouring longer hits. The contrasting high numbers of MetaPhlAn2 could be explained by its database: with MetaPhlAn2’s database missing markers for 145 of the tested viruses and having markers for diverse organisms (archaea, bacteria and eukaryotes, besides the viruses). However, it is important to mention that in the human simulations, with the true species absent in all databases, Centrifuge is the tool with the highest number of spurious taxa.

For high values of deamination and sequencing error MetaPhlAn2 and DIAMOND are exceeded by Centrifuge and Kraken2. It seems that DIAMOND and MetaPhlAn2 alignment query algorithms prevent them of increasing the number of spurious taxa as much as Centrifuge or Kraken2. DIAMOND, as well, could be more robust to noise generated by the substitutions as it depends on a protein database.

Human reads can be classified as viruses, archaea, bacteria or other eukaryotes

Ancient human reads mistakenly classified as viruses can be very problematic in studies of human remains as both viral and human reads would exhibit characteristic ancient DNA features such as fragmentation and deamination damage. To assess whether human reads had any chance of being classified as viruses if they were not properly cleaned out, we classified simulated human reads. The results show that encouragingly, most reads remain as unclassified (the species is not in the database). Nevertheless, thousands of reads are misclassified. Most of those incorrectly classified reads are assigned to a wrong species (false positive), giving the user a sense of false confidence in the classification as it is so specific. The assignments are generally hard to interpret except for a retrovirus identified by Centrifuge, Kraken2 and DIAMOND. In this case, one of the top hits (i.e. the taxa with the highest numbers of reads, see Table S3) was human endogenous retrovirus K113 (HERV-K113). As its name suggests this viral genome is integrated in the human genome (Turner et al., 2001). As a result and as we did not filter any of the simulated human data prior to classification, some simulated human reads from the reference assembly are assigned to the HERV-K113 present in the databases. The simulated reads coming from the location in the human genome where the endogenous retrovirus was integrated aligned to the HERV-K113 in the database. Note that if there had been a filtering step for human reads (such as the removal of reads that map to the reference human genome), this virus would not have been recovered (the reads would have mapped to the proviral form of HERV-K113 in the reference human genome, hence would have been filtered out).

Recommendations and future directions

In the context of ancient virome studies, the ideal classification tool is one that allows us to recover all viruses present in the studied sample but also that does not identify any spurious extra viruses. In other words, we hope to find the right trade-off between having a high number of true positives and a low number of false positives. In this study, we considered both the fraction of reads per virus that were correctly classified, as well as the number of recovered viruses and the number of spurious extra taxa with at least one read assignment.

Sequenced ancient genomes are generally incomplete and being able to correctly classify most regions in the viral genomes will be an advantage to recover ancient viruses. Given the difficulty in finding ancient viruses and in being certain they are actually ancient, every lead in the screening phase is generally followed by extensive work. These steps generally include the mapping of sequenced data to individual candidate genomes identified by classifiers (Krause-Kyora et al., 2018; Mühlemann et al., 2018a; Vågene et al., 2018; Mühlemann et al., 2020). Any classification is treated as a candidate and reducing the number of false positives is key to achieve a manageable workload, to decrease the computational resources needed and to minimise false claims.

We find that, when considering the fraction of correctly classified reads, Centrifuge and Kraken2 exhibit the highest sensitivity and precision at the species level. When including higher taxonomic ranks, DIAMOND reaches a precision close to those of Centrifuge and Kraken2; although the sensitivity remains considerably lower. Moreover, Centrifuge, Kraken2 and DIAMOND recover essentially all the simulated viruses for most simulations. Regarding Kraken2 and DIAMOND, very short fragments cannot be classified with these two classifiers when using default parameters (shorter than 35 bp in the case of Kraken2, and shorter than 45 bp for DIAMOND). We performed additional analyses by adjusting some of the parameters (using a seed of 9 amino acids for DIAMOND and reducing the k-mer size to 29 bp when building the database for Kraken2) so these tools can handle shorter sequences (see Supplemental Information 9). In summary, DIAMOND classifies very few additional reads. Moreover, for Kraken2, there is a trade-off: the number of spurious extra taxa increases. Considering all those results together, Centrifuge and Kraken2 are likely the better choice among the four tools to increase the likelihood of recovering ancient viruses.

One caveat with Centrifuge is that it outputs the largest numbers of spurious extra taxa across when the species tested is not present in its database (see Fig. 8 and Supplemental Information 8) or when the reads are highly deaminated. Hence, if the list of candidates is very wide (probably due to the extra taxa), an option to optimise computational resources for downstream analyses would be to focus on the candidates with the highest number of assignments (as we saw in the human simulations, an important proportion of extra taxa has less than 10 assigned reads). For studies with high deamination, DIAMOND could be a good choice to reduce the number of candidates for further downstream analyses (as long as the reads have the appropriate length) as its sensitivity and precision are stable (Fig. 6). We also recommend DIAMOND for fast screenings given its high Precision_s&h and as its database is smaller than Kraken2’s or Centrifuge’s and is easier to build (see Table 1 and Materials and Methods).

MetaPhlAn2 did not perform well in our analyses. While MetaPhlAn2 has advantages, such as having stable sensitivity and precision values in different conditions (Figs. 5–7), the large number of viruses that would be missed in ancient virome studies suggest it is not well-suited for such analyses. However, its small database makes this tool convenient if researchers are after specific viruses, as long as the virus of interest has a marker gene in the database or the researchers define its marker genes and add them to the database following the developer’s instructions.

For human studies, our simulations suggest that, even when the human genome is included in the database, cleaning as many human reads as possible would be necessary prior to classification to minimise the number of false positives. This would come at the expense of losing true candidates. However, we found hundreds of detected spurious extra taxa across different kingdoms within the viruses, and this could lead to false claims. To avoid having to handle so many false positives, our suggestion would be to map the data to the human genome and remove all reads that get assigned coordinates including those with low mapping qualities prior to classification using e.g. Sunbeam (Clarke et al., 2019).

The results presented here are focused on DNA viruses that infect humans. However, we expect they likely hold for viruses from other hosts as well. The reasons for the difficulty to classify some viruses for specific classifiers are mainly the specific databases used and the viral taxonomy of the tested viruses (see Supplemental Information 10). Hence, if the intention is to answer a specific biological question, i.e. to identify a given set of viruses with any of the classifiers tested here, we recommend to first make sure the viruses of interest are present in the classifier’s database. To help users in that regard, we have also included Table S2, that lists (for 60 bp) the raw number of reads per taxon for each classifier for all human DNA viruses included in this study.

To conclude, the four classifiers that we compared performed remarkably well considering they were not specifically developed to handle aDNA data and are robust to ancient DNA deamination damage. In our simulations, Centrifuge and Kraken2 outperforms the other classifiers, as Centrifuge can handle short fragments, and both of them have high sensitivity and precision values across read lengths. Moreover, for human studies, human ancient DNA contamination could lead to a very large number of false positives if human reads are not filtered properly. Finally, an area of future research would be to adapt the databases and default parameters of the classifiers to improve their precision and sensitivity values, and to increase the number of identified viruses, especially when handling short reads.

Supplemental Information

Supplemental Information 1 Table of viruses used in the study

This table includes the following information of the 233 viruses used in the study: virus name, sequence name, sequence length, type of molecule (DNA or ss-DNA), type of genome (linear or circular), sequence accession number, virus taxID and Baltimore group.

Click here for additional data file.

Supplemental Information 2 Table of reads per taxa

Raw number of reads assigned per taxa considering the accession sequence from which the reads were generated. In columns: accession numbers from which the reads were simulated. In rows: taxID to which the reads were assigned.

Click here for additional data file.

Supplemental Information 3 Extra taxa identified in the human simulations

Extra taxa identified in the human simulations. The table contains number of reads assigned to the taxon, its taxID, its name and its taxonomic rank.

Click here for additional data file.

Supplemental Information 4 Classification results for the 60 bp read set

(A) Percentage of the reads classified in each of the four categories: “correct species”, “correct higher”, “incorrect” and “unclassified” (see Fig. 1) per classifier for each viral sequence. Each bar corresponds to one of the 233 viruses selected for the simulations. Note that the viruses are not ordered the same way for each subplot. (B) Means over the 233 viruses for each classification category.

Click here for additional data file.

Supplemental Information 5 Average classification categories for all simulations (stacked bar plots)

Nine subplots showing the four classification categories (average across viral sequences) for Centrifuge (first row), Kraken2 (second row), DIAMOND (third row) and MetaPhlAn2 (fourth row) for all simulations; varying read length (first column), singe-stranded probability of deamination (second column), fold increase of substitution sequencing error (third column).

Click here for additional data file.

Supplemental Information 6 Average classification categories for all simulations (line plots)

Twelve subplots showing the four classification categories (average across simulated viral sequences) for the categories “correct species” (first row), “correct higher” (second row), “incorrect” (third row) and “unclassified” (fourth row) for all simulations; varying read length (first column), singe-stranded probability of deamination (second column), fold increase of substitution sequencing error (third column).

Click here for additional data file.

Supplemental Information 7 MALT

Click here for additional data file.

Supplemental Information 8 Archaeal, bacterial, human and viral databases for Centrifuge, Kraken2 and DIAMOND

Click here for additional data file.

Supplemental Information 9 New parameters for DIAMOND and Kraken2 to handle short reads

Click here for additional data file.

Supplemental Information 10 Unclassified viruses and viruses with high incorrect classifications

Click here for additional data file.

Supplemental Information 11 List of FTP links

Click here for additional data file.

We would like to thank the former Vital-IT platform of the Swiss Institute of Bioinformatics (SIB) and the DCSR for maintaining the computer infrastructure at UNIL. We thank Barbara Mühlemann, Nicolas Rascovan, María C. Ávila-Arcos, Diana Ivette Cruz Dávalos, Christophe Dessimoz and Christina Warinner for helpful discussions and Florian Clemente for comments on earlier versions of this manuscript. We are grateful to Arthur Gruber and two anonymous reviewers for their constructive input. The work was supported by the Swiss National Science Foundation (SFNS) and a European Research Council (ERC) grant to ASM.

Additional Information and Declarations

Competing Interests

Author Contributions

Data Availability

The authors declare there are no competing interests.

Yami Ommar Arizmendi Cárdenas conceived and designed the experiments, performed the experiments, analyzed the data, prepared figures and/or tables, authored or reviewed drafts of the paper, and approved the final draft.

Samuel Neuenschwander conceived and designed the experiments, performed the experiments, analyzed the data, prepared figures and/or tables, authored or reviewed drafts of the paper, and approved the final draft.

Anna-Sapfo Malaspinas conceived and designed the experiments, authored or reviewed drafts of the paper, and approved the final draft.

The following information was supplied regarding data availability:

The sequences and raw data are available in the Supplemental File.

The custom databases used are available in Dryad: https://doi.org/10.5061/dryad.mkkwh711w

The code is available at GitHub: https://github.com/yarizmen/benchmarking_aDNA_viruses.

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
