# Peer review of "Benchmarking metagenomics classifiers on ancient viral DNA: a simulation study"

_PeerJ, doi:10.7717/peerj.12784_

## Round 0.1 · original submission · Major Revisions

Dear Dr. Cárdenas and colleagues:


Thanks for submitting your manuscript to PeerJ. I have now received three independent reviews of your work, and as you will see, the reviewers raised some major concerns about the research. One reviewer raised serious concerns about the research and recommended rejection; however, the other two reviewers are more enthusiastic and raised some areas that need substantial improvement. I agree with the concerns of all reviewers.

I am affording you the option of revising your manuscript according to all three reviews but understand that your resubmission may be sent to at least one new reviewer for a fresh assessment (unless the reviewer recommending rejection is willing to re-review). Please address all of these concerns in your rebuttal letter.

Good luck with your revision,

-joe

·

Basic reporting

This paper reports a simulation study conducted with four bioinformatics tools widely used to taxonomically classify metagenomic reads. In this study, the authors used 120 viral sequences and simulated different features that are commonly found in ancient DNA, including different level of fragmentation, deamination damage, sequencing errors and presence of human sequences. The topic of the manuscript is very interesting and relevant, especially when considering the recent advances in ancient DNA sequencing methods.

The article is very well written and clear, the methods are based on previously described techniques, and tables and figures are relevant and properly presented.

Reading the text, several questions arose in different parts, but most of them were addressed in the subsequent sections of the manuscript.

Experimental design

In general, the work has been correctly designed and conducted. However, there are some methodological aspects of the work that are worth to be commented here (and better covered by the authors in the text):

1. Program’s parameters

The authors compare four different tools (the so-called ‘classifiers’) in regard to their performance under different conditions usually found in ancient DNA samples. It is clear that there are many variables to be assessed and this aspect restricted the possibility of testing different parameters for each program, so default parameters were used throughout the study. Nevertheless, this choice has clearly impacted the performance of these tools under the tested conditions, as is clear in the case of DIAMOND and Kraken2 (please see comments in the next item).

2. The authors chose to randomly select 120 reference viral sequences from RefSeq, including complete genomes, genome segments, and CDS. This choice, while randomly made, may have left some important viral groups out of the analysis. I would rather choose the most relevant viral genomes in terms, paying special attention to some features: (1) for multisegmented genomes, are all segments represented? (2) and; is there a good representation of all Baltimore groups? Please justify why they have chosen randomly selected sequences instead of ensuring that all viral groups and segments would be represented in the reference dataset.

Validity of the findings

I will comment below on some results that I believe should be better treated or at least discussed more deeply.

1. One aspect not raised by the authors is that given the high evolution rate of viruses, higher taxonomic levels show conservation restricted to some relatively short regions, those that are under strong selection pressure. Conversely, lower levels (more specific) tend to show longer conserved regions certainly impact the overall performance of tools that use different approaches to classify sequences. More divergent sequences (belonging to higher levels) probably show overall higher heterogeneity of kmers, while still maintaining conserved stretches that can efficiently be detected by a sequence aligner such as DIAMOND, thus explaining why this program shines in the ‘correct higher’ category. This aspect should be better covered in the Discussion.


2. It would be very interesting if the authors could provide some clues on why some viruses have shown poorer classification for some of the tested programs. For instance, would they show different compositional biases, do they belong to viral groups that show higher evolutionary rates (e.g. RNA viruses), etc.

3. Regarding multisegmented genomes, let’s take as an example the case of the Bunyavirales (e.g. the genus Orthobunyavirus). These viruses show a tripartite genome, with segments coding for the RDRP (L segment), the nucleoprotein (S segment) and Gn-Gc glycoprotein (N segment). It is well known that the L segment is much more conserved than the S and N segments. I would be very interesting to assess how variable is the performance of each classifier for these viruses, taking into account the different segments. This is an example on how interesting and appropriate a rational choice of the viral sequences used in the reference dataset would be.

4. Another important aspect regards retroviruses and HERVs (human endogenous retroviruses). Considering that the human genome does contain a large collection of these elements, which show cross-similarity to exogenous retroviruses, this condition must have strongly impacted the performance of the classifiers, especially when considering that there was no specific methodological step to deal with this problem. This topic should be much better explained and discussed than presented in the Discussion (lines 658-662).

5. A noteworthy result is the finding that DIAMOND shows relatively poor performance for the correct species assignment while displaying much better results for higher taxonomic levels (lines 319-323). Using default parameters was certainly a limiting factor for a good performance of DIAMOND and Kraken2 in some tests. Having admitted this fact (lines 589-596), the authors should refrain from being so emphatic in their conclusions:

“Nevertheless, short reads cannot be handled by either Kraken2 or 683 DIAMOND” (lines 682-683).

These programs certainly can deal with short reads, but require using the most appropriate parameters rather than the default parameters.

Additional comments

No comment.

Reviewer 2 ·

Basic reporting

In this study, entitled “Benchmarking metagenomics classifiers on ancient viral DNA: a simulation study”, Arizmendi Cárdenas and colleagues explore the performance of four metagenomic classifiers for detecting ancient viral reads, to assess their potential for being used in future palaeomicrobiological studies. After a number of tests, assessing the effect of potential aDNA substitutions, read length, and human DNA within specimens, the authors conclude that out of the tools Centrifuge, Kraken2, DIAMOND and MetaPhlAn2, Centrifuge would be the best performing tool and most applicable for short-read ancient DNA data. The paper is written well and the scientific goals of the study are made clear throughout the introduction. Regarding the study’s background, one point of improvement within the introduction concerns proper citation of relevant literature. The authors should mention previous attempts on comparison of metagenomic classifiers for ancient pathogen detection as, currently, no such citation is provided. Moreover, certain publications could be re-considered, as they are inappropriate as references to some of the discussed concepts. For more specific comments, refer to the sections below.

Experimental design

Importantly, in terms of the investigation’s scope, while such a study is appreciated and needed within the community of palaeomicrobiologists, as so far similar assessments only exist for ancient bacterial sequences, there are several concerns that arise about the study’s design and, as a result, outcome. Below you can find a number of general suggestions, as well as specific comments, that should be addressed in a new version of the paper.

Validity of the findings

no comment

Additional comments

General comments:

1. One of the classifiers chosen to be tested by the authors is DIAMOND. DIAMOND is a taxonomic binning tool operating under an amino acid database. The developers of DIAMOND have more recently released the tool MALT, whose concept of taxonomic binning is very similar to DIAMOND, but operating under a nucleotide reference database. MALT (Vagene et al., 2018) has been often used to detect bacterial and viral sequences from ancient human remains and has been developed further within the pipeline HOPS (Hübler, Key et al., 2019) to tailor the postprocessing of results for aDNA-aware assignments. Given the abovementioned characteristics, why do the authors choose to include DIAMOND in this comparative assessment study instead of MALT? I suggest MALT be considered as a more appropriate tool for this comparison. This is particularly important especially in light of the author’s finding of DIAMOND’s poor performance in classifying 30bp reads, which are typical of endogenous ancient DNA.

2. The authors mention within the methods section that viral sequence datasets were generated to contain 10x representation of each virus. Such a viral coverage goes beyond what is expected from ancient metagenomic samples, especially for shallow-shotgun sequenced libraries that are usually used for pathogen screening. For example, a recent study by Muehlemann et al., 2018, on HBV evolution showed that as little as two viral reads were detected in shotgun-sequenced ancient specimens prior to hybridization capture. Therefore, I suggest that tests be performed with a significantly lower representation of target viral reads within the simulated datasets, similar to previously presented results of ancient DNA studies.

3. Moreover, the highly dominant environmental background, which is typical in ancient metagenomic specimens, is not included within the simulations performed in this study. Environmental contamination is one or the most typical characteristics of ancient metagenomic specimens, and one of the main complicating factors associated with ancient pathogen identification by metagenomic classifiers (discussed in detail in the review by Warriner et al., 2017). I suggest that tests be altered to better reflect true metagenomic data deriving from ancient specimens. Moreover, I suggest that target viral reads occupy a small percentage within those datasets, <0.1%, to reflect typical ancient pathogen endogenous DNA estimates. Such simulations can be performed within gargammel that is already used by the authors to simulate aDNA damage in this study.

4. The authors describe that “A total of 120 reference viral sequences were randomly selected from the viral genomic NCBI RefSeq database”. Here, the authors should provide the entire list of viral sequences that went into this simulation study. Moreover, the authors should elaborate on the reasons of why a random selection process was used, whether viral genomes previously detected by aDNA studies were included in their list, and whether the list was restricted to DNA viruses only or also contained human-infecting RNA viruses.

5. Regarding the incorrect read assignments (Figure 4C, 5D, 6D, 7D) it would be critical to investigate whether these tend to always be from specific viral sequences that appear to be more difficult to classify than others, for example from viral taxa with high genomic diversity. Moreover, where do the “incorrectly assigned” reads end up? Do these end up on higher taxonomic nodes because of their conservation among several taxa? Specifically, it would be important to check whether “incorrect assignments” are most often seen on over-represented taxa within the reference database. Results from this test could inform on the importance of balancing database entries for future studies.

6. The authors restrict their assignments to read thresholds higher or equal to 50 reads, in order to avoid spurious read assignments. However, recent studies have shown the detection of ancient DNA viruses with fewer than 10 reads, both for larger DNA viruses (e.g. Variola, see Muhlemann et al., Science 2020) as well as for DNA viruses with shorter genomes (e.g. HBV, see Muhlemann et al., Nature, 2018). How are the present findings corroborated in light of the recently published literature?

Specific comments:

- The authors state in their Introduction “The availability of aDNA has opened a unique window into the past allowing for instance to pinpoint the origin of a pathogen (Taubenberger et al. 2005; Mühlemann, Jones, et al. 2018; Rascovan et al. 2019)”. This sentence should be altered since none of the cited papers has effectively pinpointed the origins of any of the studied pathogens. Within this context, it might be worth to mention the oldest recovered viral genome, which to data is the one presented by Krause-Kyora et al., 2018.

- In order to reflect the potential of ancient pathogen studies, the authors state in their Introduction that such data are able to “…to characterize past epidemics and the prevalence of pathogens across time (Mühlemann, Jones, et al. 2018)”. This sentence and citation should be revised, as the cited research does not reflect the characterisation of a pandemic, and “prevalence” is an epidemiological measure that cannot be estimated using ancient DNA due to inherent data biases.

- “Additionally, HBV’s prevalence across time was found to vary from the Bronze Age to Middle Ages across space (Mühlemann, Jones, et al. 2018).” Please correct the statement on “prevalence” as discussed above.

Reviewer 3 ·

Basic reporting

The manuscript from Arizmendi Cárdenas et al. presents a benchmarking study aiming at identifying the most suitable tools for the detection and taxonomic classification of viral sequences in ancient DNA datasets. In the study, they test four softwares (Kraken2, Centrifuge, DIAMOND and Metaphlan2) with simulated sequencing data with aDNA degradation signatures produced from 120 randomly chosen viral genomes. They present different metrics to assess their capacity to detect these viruses (precision and sensitivity) as well as their false positive and false negative rates. Although the question behind the study is valid, I find that the work has several major flaws that must be addressed.

Major points:

- The study types for which the benchmarking is tailored is not clear.
The manuscript is presented as aiming at detecting any kind of virus in ancient DNA data (RNA and DNA based), and the focus continuously change along the manuscript (i.e., in some sections they seem to be focused on ancient human pathogens, in others in animal infecting viruses, etc.). However, depending on the kind of virus that are searched in a study the appropriate methods for the analysis will be different. In that sense, it is not possible to benchmark methods universally, to deal with every type of study. In that sense, the manuscript does not properly test the methods for different kinds of situations and does not discuss either in which kind of situations each of the tested methods could work better. For instance, how good these classifiers are for detecting known human infecting viruses, to detect animal infecting viruses, to detect known phages from the human microbiome, or phages in general, or viruses in a non-human ancient sample (animal, plant, sediments, etc.). These categories I am listing are just illustrative (I don’t pretend you to use them as such), I am just saying that depending on the goal of the project, the kind of viruses that one would like to detect are different and as well as the appropriate methods to detect them.


- I have several concerns with the testing dataset of 120 viral genomes.
First, the list of genomes used in this dataset is not given. Second, it is neither explained how was the random sampling done. One would expect that certain viral groups are overrepresented respect to others. Third, and related to my previous point, the selection of the viral genomes should respond to an a priori question, i.e., which kind of viruses would you be interested in searching in an aDNA study? From my point of view the choice of a random set of genomes from all known viruses does not reflect the kind of things that are investigated in most aDNA studies. Fourth, I don’t understand the choice of including genome segments and coding sequences… Most likely for such cases there will be a whole genome of the same species in the database that should be used instead. The fragmented nature of aDNA data should be modeled by generating very few reads from these genomes, not by using incomplete sequences as starting point to generate simulated reads.

- The tested methods and the databases used were not correctly chosen.
First of all, MetaPhlan2 was not a suitable method if any virus in RefSeq was to be tested. Viruses don’t have universal marker genes as cellular organisms, so it is not possible to use it for viral detection. The authors themselves show that two thirds of the tested viruses could not be detected, which was absolutely expected even before running any test. For the case of DIAMOND, the only way to directly test sensitivity, precision and incorrect classifications compared to Kraken2 and Centrifuge was to use the exact same database as in these methods (i.e., either all with the full RefSeq DB, or all with a viral DB). I think the full RefSeq DB for the three is the best choice, because it can reduce false positives. And coming back to my first point of the type of study, if we were trying to characterize an environmental virome, where most viral genomes won’t be likely present in the DB, DIAMOND would be a better choice, because it has the highest precision (Fig 4A) and its protein level analyses with SEED extension and LCA approach will allow classifying genomes that are more divergent from those in databases with much better resolution than k-mer based methods (which will fail at classifying a certain taxon if it’s full of synonymous mutations). K-mer based approaches on the other hand would be a better choice when a specific well-known species is searched (e.g., a DNA viral pathogen such as HBV or variola), although in those cases just mapping reads (with bwa or bowtie) to the reference viral genomes would be even better. On the other hand, if you were looking for human viruses related but not identical to those in databases, you should have chosen viral genomes for such case.
For instance, if you were using a RefSeq database from before 2018-19, you could have tested Sars-CoV-2 among your 120 test viruses, which would inform on the best method for this kind of cases. This is relevant for aDNA, because we may be searching for viral species from certain viral pathogen groups that were infecting populations in the past and are now gone.

- Simulating reads covering viral genomes at 10X does not really represent what would happen in an aDNA study. Indeed, the most typical problem is that there are only few reads of a certain microbe (e.g., mean coverage at 1X or less), so we must identify their presence with highly fragmented data. Hence, I think that different coverages should have been tested. I believe that at very low coverage, the effect of deamination and sequencing error may become more evident.

- The manuscript is unnecessarily long.
The manuscript has 22 pages, where the same content could have been described in 5 or 6 pages. There are too many sections in results and discussion that could be collapsed, and the authors repeat themselves a lot (e.g., the discussion section repeat a lot what has been already said in the results), while not discussing properly some relevant points. I think that Results and Discussions should be presented together. For instance, I was having concerns or thoughts about certain results, which were finally addressed in the discussion many pages after.


Altogether, I think that the manuscript needs still a lot of work and to address these major points before being considered for publication.

Experimental design

There are some major problems in the experimental design, as listed in the Basic reporting and the general comments to authors. Some of these include:

- Issues in defining the scope of the work (i.e., which kind of aDNA studies and kind of viruses are tailored)
- Issues with the choice of viral genomes for the tests, and the coverage used for the simulated reads
- Issues with the selected methods and the databases used for these methods, that complicate the comparison of results

Validity of the findings

Because of these flaws in the experimental design and definition of the scope, the findings are not very conclusive, and it would be hard anyway to drive any conclusion. There are also issues in how results are discussed (e.g., in which case each method would be better).

Additional comments

Then I also made several comments while reading the manuscript, which I list in order of appearance:


Abstract: There is a very frequent use of the present tense. I believe the past tense would be more accurate, because you are talking about the work you did.

Line 32-33: Do you have any reference to support this: “making the identification of ancient viral genomes particularly challenging”

Intro: I think the paragraph from L67 to L84 could be reduced to just one sentence to mention that recovery of ancient viral genomes is possible, then the citations, but for the scope of the article, the particular examples are not relevant.

Lines 96-106: This paragraph about the characteristics of aDNA can be also drastically reduced to just the main key points. For the sake of this work, the only important are the C-to-T/G-to-A transitions, the shortage of read lengths and maybe something on the quality.

Line 180 and related to major point 2): I don’t understand the rationale behind the choice of the 120 genomes. I got several questions while reading this. For instance, the world virosphere is extremely large, and highly under sampled, mostly if we consider phages. However, the human virosphere is much more characterized, and especially if we consider the human infecting viruses, and even more if we consider the pathogenic ones. So, which of these levels are your main targets? Your introduction seems to point towards an interest in human pathogenic viruses, but your dataset of 120, although not shown, seems to be for any virus.
In that sense, rather than just 120 random viral genomes based only in their genomic length, I would have taken random representatives of these heterogenous groups. When it comes for aDNA data, it won’t be the same to classify ancient phages in the microbiome, than ancient human infecting viruses. Moreover, I did not find the list of the 120 viral genomes anywhere (maybe I missed it). This should be shared, at least as a supplementary table. I wonder at which extent this random choice is not biased by the fact that certain viral groups are far more represented in NCBI database than others (e.g., today there are more than 500k genomes of Sars-CoV-2 alone).
Also, RNA viruses are in principle less relevant since they are generally not tackled in aDNA studies, since no one is really sequencing ancient cDNA. You can test with few, in the case anyone tries it in the future, but we can all agree that this is not a common practice in the field.
Finally, having so many whole genomes in databases, I don’t see the point of including the segments and CDS. What do you gain from that? This does not reflect the kind of data you’ll get from an ancient sample, and for most cases, you will have a close related genome in database you could use instead.


Line 200-201: You should mention here that other qShift values were also tested, which is mentioned after.

Lines 203-206: What’s the rationale behind the choice of 10X for the reads. In an aDNA study such a high coverage is rarely the case. Moreover, wouldn’t you want to know whether the virus can be correctly detected and classified when you have only few reads from it?

Lines 219-224: Why DIAMOND was not used with the same or similar RefSeq DB as Kraken and Centrifuge? By just using the viral genomes, you can get false positives (sequences that otherwise would be misclassified) and you can also overlook the false negatives it could give. Indeed, when looking at figure 4A, Kraken and Centrifuge show practically identical results and I just wonder if results with DIAMOND would not improve to some extent if we were using the same database as with Kraken and Centrifuge.

Lines 225-228: I don’t quite see the point of including MetaPhlan2. The analyses presented in the manuscript are tailored at detecting any kind of virus, not some particular groups that have marker genes. There are not many conserved genes in viruses, hence only a very limited group of viruses can be detected (those groups that are better characterized and contain marker genes). In that sense I am not sure at which extent it can be compared to the other methods for the viral dataset you used for benchmarking.

Figure 1: In the “molecule” field for the database, you should use “nucleotide”, since you are also including RNA viruses in the database.

Figure 3: Here we can see that substitutions due to deamination do not reach the average mean of 1 per 600bp with any deamination probability tested. I wonder if this is what we would normally expect at maximum in an ancient viral DNA, based on other literature examples. From Figure 3B, it seems that sequencing error would exceed the number of mutations due to deamination at qshift 11. This is reflected in Figure 6 and 7, where results barely change for deamination (probably because there are not really many mutations introduced) and do decrease a bit due to sequencing errors. I would nevertheless mention upfront which are the viruses that are no longer classified in each case to try to understand which are the features or cases in which each software is most sensitive. Some mentions to this are given, but way after in the text.

Figure 4B: You should also mention upfront which are the viruses that are not detected (i.e., there is one virus missing from Kraken2, Centrifuge and DIAMOND, but it is not the same one for DIAMOND. The identify of these viruses should be mentioned before in the text. Also, how would you explain the decrease in “# of viruses detected” on DIAMOND when increasing the reads size threshold?

Line 296: Human simulations.
I believe that the reads that fail to be removed from aDNA data are not at any random region in the human genome. In that sense, instead of testing reads from any region of the human genome, I would have used human sequences that failed to map into the human reference genome, which are in fact the ones that are likely to escape the human cleaning step.

Figure S2: Both DIAMOND and Metaphlan2 show a high number of reads non classified. This is certainly expected, since DIAMOND would only use information from protein coding genes, and Metaphlan2 only marker genes. Any read not mapping into these subsets, will therefore not be classified. However, this does not mean the method is less useful depending on what the goal is (e.g., retrieving all reads from a detected viral pathogen or getting an idea of the whole viral diversity in the sample, etc.). Maybe a way to overcome this issue would be to additionally calculate sensitivity and precision based on total classified reads, not only on total reads (i.e. not counting the unclassified). This analysis will certainly favor DIAMOND followed by Metaphlan2, because they have less proportion of “incorrect” reads (as you mention also in L344-345).

Line 329-331: Would the proportion on “incorrect” reads change if you were using the whole RefSeq DB for DIAMOND? Maybe the fact that there is a higher proportion of “incorrect” in Kraken2 and Centrifuge is just related to the larger database that it was used with the latter methods (e.g., viral sequences misclassified as some bacteria or eukaryote that were not present in DIAMOND DB). In that sense, which kind of viruses were exceeding the 60% “incorrect”?

Lines 373-393: The number of spurious taxa in this paragraph are confounded by the fact that a different database was used for DIAMOND. L390-393: these thresholds are nearly impossible to establish universally because there is no way to predict how well a viral DNA will be conserved in an ancient sample.

Line 403-404: As it is also acknowledged by the authors, this was simply because of an error when building the databases or when choosing the read lengths for the tests. If you would have used a k-mer size of 27 to build the database, it would have worked, but then the other results may have changed with shorter k-mer sizes. Same case for DIAMOND. This is not a result to discuss in the article in the way you did, this is simply an error in the experimental design. Again, the comment remains the same, things should be compared with equivalent parameters every time is possible.

Lines 534-540: This result is not discussed. It is probably due to the presence of human sequences in these RefSeq genomes? Low-complexity sequences? What?

The Discussion is too overlapping with the results section. I would recommend to present results and discussion together to save space and also because some of the results need to be discussed while describing them.

Lines 577-596: As I said before, it is wrong to say that Kraken cannot lead with short reads. You should had built the database with a k-mer size according with the read lengths you were going to test.

Line 610-612: it is because many viral groups don’t have a marker gene.

Line 613-621: The discussion about DIAMOND is somehow wrong. Proteins have less variability than DNA, because the genetic code has many synonymous, so different DNA sequences can give the same protein. Also, you should discuss here in which cases DIAMOND could be more suitable.

Lines 626-629: I don’t think that deamination at 0.5 is comparable to qShift at -11. While deamination 0.5 may add one to no mutations, qShift -11 adds more than one. The results obtained is just a consequence of using a very high error rate, and a modest deamination probability.

Lines 630-633: I don’t understand this conclusion. It looks contradictory to what you said right before.

Lines 634-648: As I mentioned before, comparing Kraken2 and Centrifuge on all RefSeq, while DIAMOND in only viruses is a mistake that turns them hardly comparable.

Lines 673-677: Here you are mixing things, I think. If you were looking for specific ancient viral pathogens species (as in both papers cited here), you can simply map with bwa or bowtie on a database of known viral pathogens (an analysis that surprisingly you did not benchmark). So, this comes back to my main comment at the beginning, depending on the type of study, different strategies could be better. Here you are just focusing on one, so which case are you interested in?

Line 697-700: As I said before, most people remove the human genome prior to the analysis of microbes, hence you should perform the analyses with the human reads that remain after mapping to the reference (there is always a fraction that get not mapped).

Lines 705-707: This is a surprising statement to conclude the article! The benchmarking presented in this paper is tailored to which cases? If we have to benchmark again for our particular studies anyway, what’s the purpose of this article then?

---

## Round 0.2 · accepted · Accept

Dear Dr. Cárdenas and colleagues:

Thanks for revising your manuscript based on the concerns raised by the reviewers. I now believe that your manuscript is suitable for publication. Congratulations! I look forward to seeing this work in print, and I anticipate it being an important resource for groups studying approaches to characterizing and analyzing ancient viral DNA and genomes. Thanks again for choosing PeerJ to publish such important work.

Best,

-joe

·

Basic reporting

The authors submitted an extensively revised article and answered all my questions satisfactorily. Many experiments were redone and some new ones were included. In addition, the text was also greatly changed in order to meet all the points raised. I consider the article to be suitable for publication.

Experimental design

All points raised in my previous review have been addressed.

Validity of the findings

Changes made to the text and new experiments are in accordance with the findings and their interpretations.